# Urbanization Level in Chinese Counties: Imbalance Pattern and Driving Force

**Baifa Zhang** [1,2], **Jing Zhang** [1] **and Changhong Miao** [1,2,*]

1 Key Research Institute of Yellow River Civilization and Sustainable Development & Collaborative Innovation Center on Yellow River Civilization Jointly Built by Henan Province and Ministry of Education, Henan University, Kaifeng 475001, China; zbf@henu.edu.cn (B.Z.); 104753181181@vip.henu.edu.cn (J.Z.)
2 College of Geography and Environmental Science, Henan University, Kaifeng 475004, China
* Correspondence: chhmiao@henu.edu.cn

**Abstract:** Urbanization level is a key indicator for socioeconomic development and policy making, but the measurement data and methods need to be discussed further due to the limitation of a single index and the availability and accuracy of statistical data. China is urbanizing rapidly, but the urbanization level at the county scale remains a mystery due to its complexity and lack of unified and effective measurement indicators. In this paper, we proposed a new urbanization index to measure the Chinese urbanization level at the county scale by integrating population, land, and economic factors; by fusing remote sensing data and traditional demographic data, we investigated the multi-dimensional unbalanced development patterns and the driving mechanism from 1995 to 2015. Results indicate that: The average comprehensive urbanization level at the Chinese county scale has increased from 31.06% in 1995 to 45.23% in 2015, and the urbanization level in the permanent population may overestimate China's urbanization process. There were significant but different spatial and temporal dynamic patterns in population, land, and economic levels as well as at a comprehensive urbanization level. The comprehensive urbanization level shows the pattern of being high in the south-east and low in the north-west, divided by "Hu line". The urbanization of registered populations presents high in the northern border and the eastern coastal areas, which is further strengthened over time. Economic urbanization based on lighting data presents high in the east and low in the west. Land urbanization based on remote sensing data shows high in the south and low in the north. The registered population urbanization level is lower than economic and land urbanization. County urbanization was driven by large population size, reasonable industrial structure, and strong government capacity; 38% and 59% of urbanization levels can be regarded as the key nodes of the urbanization process. When the urbanization rate is lower than 38%, the secondary industry plays a strong role in powering urbanization; when the urbanization rate is higher than 38% but less than 59%, the promotion effect of the tertiary industry is more obvious, and the secondary industry is gradually weakened. When the urbanization rate exceeds 59%, the tertiary industry becomes the major driver.

**Keywords:** county urbanization; population urbanization; land urbanization; economic urbanization; comprehensive urbanization; China





## 1. Introduction

Urbanization has developed rapidly in the 20th and 21st centuries and has been regarded as an important development strategy [1]. With increasing global urbanization, the Earth is gradually becoming an urban planet [2,3]. The proportion of the global urban population has increased from 33% in 1950 to 55% in 2018, and urban land expanded at twice the rate of population growth [4]. Rapid urbanization in China requires more attention from the world. Stiglitz, a Nobel laureate in economics, once predicted that China's urbanization was one of the two major engines driving world economic development in the

21st century [5,6]. However, due to the complexity and comprehensiveness of urbanization, the development and change of urbanization have always been a hot issue for governments, scholars, and the public [7,8]. So, how to measure the level of urbanization development accurately has strong theoretical and practical significance [9].

The measurement of urbanization level is particularly important in the stage of rapid urbanization development. Northam believes that urbanization development presents an "s"-shaped curve and can be divided into three stages and that the rapid urbanization stage is when the urbanization level is between 25% and 70% [10]. At this stage, one of the research objectives in China and abroad is how to measure the level of urbanization scientifically. Right now, China is at this stage [7–9]. As the fastest growing economy in the world, China's urban population increased from 200 million in 1980 to 800 million in 2018 [11]. In 2019, the urbanization rate in China exceeded 60%, and the urbanization rate of household registration reached 44.38% [12]. However, scholars have certain disputes over the current level of urbanization [13] and believe that the current level of urbanization is overestimated [14,15]. Therefore, how to evaluate the level of urbanization in China scientifically is an important issue [16–18].

Through a literature review, the selection of indicators is an important issue in the study of urbanization. The existing studies are mainly based on a single indicator or composite indicator system, which cannot reflect the urbanization process comprehensively and truly. In terms of a single indicator, the research on measuring the level of urbanization based on population data is the most used and the most representative. As early as the 1930s, the "Hu Huanyong Line" showed the characteristics of the spatial pattern of China's population was extremely dense in the east and sparse in the west. Although these findings are from a population density map, they also reflects the differentiation of urbanization levels to a certain extent [19]. Since then, the research on population urbanization has gradually increased [20–22]. Moreover, there is also much research on India, Central Asia, and other countries and regions using population data [23–25]. With the intensification of population mobility, the measure data also changes [26]. Qi believed that the urbanization rate of registered populations could more scientifically reflect the quality of China's population urbanization, so he used the data of the sixth population census to measure the urbanization rate of the permanent resident population and registered population at the county and city scales in China [27].

It is also the content of single index research to analyze the urbanization process from the dimension of land use. With the rapid development of urbanization, it is not only manifested in the increase of urban population but also manifested in the disorderly expansion of out of control urban construction land in space. Important resources such as arable land and water are overconsumed, the environment is seriously polluted, and urban infrastructure is greatly wasted. Therefore, in 2007, Mr. Lu proposed to use land urbanization for the first time in China and pointed out that the speed of land urbanization was too fast and much faster than population urbanization [28]. The research on land urbanization has become a hot spot. Some scholars [29–31] proposed that the land-use area of urban built-up areas in a country or region reflects the local urbanization level. Moreover, other scholars believed that the proportion of industrial and mining land in urban areas could better reflect the process of land urbanization and propose that the ratio of urban and rural construction land to urban construction land can be used to express the land urbanization rate [32–34]. Furthermore, Asabere pointed out that the spatial change process caused by urbanization can be better revealed by analyzing the change in regional land use and cover quantity, quality, and structure [35]. Vogler conducted a case study of the United States and showed that per capita land change could be a new spatial index for measuring urbanization [36]. In addition, the use of per capita GDP, the proportion of secondary and tertiary output values and composite economic indicators to measure economic urbanization, is also one of the focuses of urbanization research [37–41].

Although the single index measure has strong brevity, it cannot fully reveal the connotation of urbanization, and there are certain errors in judging the development

stage of urbanization [42–44]. To solve these problems, scholars have constructed a multi-standard comprehensive evaluation index system to measure the level of urbanization (i.e., population and land, population and economy, land and economy, population-land and economy) [5,6,45–52]. Furthermore, although the multi-dimensional index system is more scientific than a single index in measuring urbanization, the data source of the multi-dimensional index system is still mainly the traditional social and economic statistical data, while the traditional data has many limitations (data quality is relatively low; temporal continuity is weak; spatial scale is limited, etc.) [1,53]. With the digital wave sweeping the world, using remote sensing images to do urbanization research has gradually become a new research direction [54,55], and the fusion of social economy, satellite remote sensing, social media, and other multi-source data is the mainstream trend of future research [56,57]. Among them, lighting data is the most widely used and provides a good opportunity to monitor annual urbanization activities. The light data provided by Operational Line-scan System (OLS) on the Defense Meteorological Satellite Program (DMSP) can detect city lights with a low-light detecting capability [58–60]. Due to the advantages of rich historical archived data and wide spatial coverage [61,62], lighting data has been widely used in socio-economic estimation [63,64] and urban agglomeration development [65]. For example, Sutton et al. [66] estimated the global human population using the statistical relationship between nighttime lighted areas and the urban population.

In order to solve the limitation of a single index and the data traditionally attributed to the composite index system based on the connotation of urbanization, we combine multi-dimensional (population, land, and economy) and multi-source data (population statistics, land use, and night light), and propose a new urbanization measurement indicator—comprehensive urbanization. Meanwhile, China is urbanizing rapidly, but the urbanization level at the county scale remains a mystery due to the complexity and lack of unified and effective measurement indicators. Therefore, we take China's counties as the research object and use the newly constructed comprehensive urbanization index to answer three questions: What is the level of county urbanization in China from 1995–2015? What are the characteristics of geographical imbalance? What are the driving forces and mechanisms? By answering these questions, we try to provide new ideas and methods for urbanization measurement, and we try our best to show the real and comprehensive process of county urbanization in China so as to provide guidance for future development of urbanization.

## 2. Research Methods and Data Sources

### 2.1. Method of Urbanization Measurement

Population urbanization: Human behavior is generally regarded as one of the direct driving factors affecting and changing urbanization. Non-agricultural population refers to the population engaged in the secondary and tertiary industries and the part of the population supported by them. Due to the small population flow in the early years, the high urban-rural dual barrier and the large difference in treatment between the non-agricultural population and agricultural population, the ratio of the non-agricultural population to the total population can better reflect the population urbanization rate in China. After 2000, the ratio of the permanent urban population to total permanent population was used to represent the urbanization rate, which was better reduced to the error effect caused by population flow. However, the permanent population statistics include the floating population that has not really achieved the process of settling down [21,67]. Therefore, after considering the availability of data, this paper uses the urbanization rate of the registered residence population to show the urbanization development pattern of the county population in China. The urban rate of registered population is the ratio of the non-agricultural population to the total registered population of a county, explained in Formula (1).

$$\text{Urban}_{\text{pop}} = \frac{\text{pop}_{\text{non\_agri}}}{\text{pop}_{\text{all}}} *100\% \tag{1}$$

In Formula (1), $Urban_{pop}$ represents the urbanization rate of the registered population; $pop_{non\_agri}$ represents the non-agricultural population of a county; and $pop_{all}$ represents the total registered population of the county.

Land urbanization: Land is the carrier of urbanization and the basis of human activities. Land urbanization is the process of transforming agricultural land into non-agricultural land. In China, due to the existence of ecological protection red line, permanent basic farmland protection red line, and urban development boundary, all the areas of a region cannot be completely transformed into urban land. At the same time, it is reasonable to use the land urbanization index and land non-agricultural index to construct the county land urbanization rate [34]. Therefore, this paper uses the ratio of urban land, industrial and mining land, and transportation land to the total size of urban and rural construction land to reflect the land urbanization Formula (2).

$$Urban_{land} = \frac{ul + il + tl}{ul + il + tl + rl} * 100\% \tag{2}$$

In Formula (2), $Urban_{land}$ represents land urbanization rate; ul represents urban land area; il represents industrial and mining land area; tl represents transportation land area; and rl represents the rural residential land area.

Economic urbanization: Industry plays a crucial role as the internal driving force of urbanization [68]. Due to the close relationship between lighting data and economic development [69,70], light data provided a valuable data source for elucidating the dynamics of China's urbanization [71–73]. Referring to land urbanization and population urbanization models, "economic activity area" and "economic activity intensity carried by land" are incorporated into the economic urbanization model. Based on the existing studies, this paper constructs a model reflecting economic urbanization from two aspects of night light: (a) the intensity attribute of regional average light; (b) the area attribute of regional lighting [74–77]. Finally, the model of economic urbanization is constructed by combining the two indexes in a linear way, shown in Formula (3).

$$Urban_{eco} = \left( \frac{TDN}{N * 63} * W_1 + \frac{Area_N}{Area} * W_2 \right) * 100\% \tag{3}$$

In Formula (3), $Urban_{eco}$ represents the economic urbanization rate; TDN represents the total value of bright elements in a certain county; N represents the number of bright elements; and 63 represents the maximum value of a single bright element. The first formula of the model represents the intensity attribute of regional average light, that is, the ratio of the actual total value of the light element in a county to the theoretical maximum value of the light element in that county. $Area_N$ refers to the total area with a pixel value greater than 0 in a region, and Area refers to the area of a county. The second formula in the model represents the area attribute of regional lighting, the ratio of the total area of all light elements in the county to the total area of the county. By verifying the weight combination, it is finally determined that $W_1$ is 0.5 and $W_2$ is also 0.5.

Comprehensive urbanization: The urban-rural system contains three important elements: population, land, and industry [41]. Urbanization is usually used to refer to three related but different processes, including the transformation of an agricultural population into a non-agricultural population, an agricultural region into a non-agricultural region, and agricultural activity into non-agricultural activity [42,78]. Therefore, this paper studies urbanization from the perspective of population, land, and industry [47]. By comprehensively considering population urbanization, land urbanization, and economic urbanization, the comprehensive urbanization rate is constructed by assigning the three weights, respectively, expressed in Formula (4).

$$T = \alpha * Urban_{pop} + \beta * Urban_{land} + X * Urabn_{eco} \tag{4}$$

In Formula (4), T represents the comprehensive urbanization rate. A, β, and X are the undetermined coefficients of population urbanization rate, land urbanization rate, and economic urbanization rate, respectively. Since population, land, and economic urbanization are equally important, the three are assigned the same weights in this paper, which are all 1/3.

## 2.2. Fixed Effects Panel Regression Model

This paper selects four indicators from three aspects of population size, industrial structure, and government capacity, namely population density, the proportion of primary industry in GDP, secondary and tertiary industrial structure, and the level of fiscal transfer payments (Table 1). Specifically, first, as the core element of urbanization, population size provides basic conditions for the development of urbanization. Secondly, economic development is an important driving force of urbanization, and industrial structure is an important representation of economic development. As a largely agricultural country, the development of primary industry may restrict the development of urbanization in China. At the same time, the tertiary industry has a stronger employment absorption capacity than the secondary industry, and its development may promote the rapid increase of the urbanization rate. Therefore, the proportion of primary industry in GDP and the structure of secondary and tertiary industries are selected as the indicators of industrial structure. Thirdly, China has a unique administrative system, and the government has played a strong role in the development of urbanization. The level of fiscal transfer payments can better characterize the government's ability. The panel regression model is constructed based on the selection of variables, and the expression is as follows:

$$T_{it} = \alpha_{i,t} + \beta_1 \ln pd_{i,t} + \beta_2 ppi_{i,t} + \beta_3 tos_{i,t} + \beta_4 gov_{i,t} + \varepsilon_{it} \tag{5}$$

**Table 1.** Index selection of influencing factors of county comprehensive urbanization.

| Primary Index | Secondary Index | Indicator Description |
|---|---|---|
| Population size | Pd | Population size/county area |
| Industrial structure | Ppi | Primary industry value/GDP |
| | Tos | Tertiary industry value/Second industry value |
| Government capacity | Gov | (Expenditure-Revenue)/GDP |

In Formula (5), T stands for comprehensive urbanization; pd is the population density; ppi is the proportion of primary industry in GDP; Tos represents secondary and tertiary industrial structure; and gov represents the level of fiscal transfer payments (Table 1). $\alpha_{i,t}$ is the individual fixed effect; $\beta_1 \sim \beta_4$ is the regression coefficient of each variable; $\varepsilon_{i,t}$ is the random disturbance term; and i and t represent region and time, respectively.

In order to ensure the accuracy of the regression results, the model is processed and tested as follows: (a) In order to ensure the data stability and weaken the influence of collinearity and heteroscedasticity of the sequence on the estimation results, some indexes of the model are treated with logarithm; (b) through F-test and Hausman test, it is found that the chi-square *p* value in the model is less than 0.05. Therefore, this paper selects the fixed effect as the estimation method of the model.

## 2.3. Threshold Regression Model

We take population urbanization, land urbanization, and economic urbanization as threshold variables to test the threshold effect of Tos on comprehensive urbanization. Then determine how secondary industry and tertiary industry contribute to urbanization at different stages of urbanization.

$$Y_{it} = \beta_1 x_{I,t} I(I \leq \gamma_1) + \beta_2 I_{i,t} I(\gamma_1 < I \leq \gamma_2) + I x_{i,t} I\left(I < g_{i,t} \leq \gamma_n\right) + \beta_4 coltrol_{i,t} + \varepsilon_{it} \tag{6}$$

In Formula (6), i stands for the county, and t stands for the year; $Y_{it}$ is the dependent variable (comprehensive urbanization), $x_{it}$ is the explanatory variable (tos); $\beta_1$, $\beta_2$, and $\beta_3$ are the corresponding coefficient variables; $g_{it}$ is the threshold variable; population urbanization rate, land urbanization rate, and economic urbanization rate are selected as the threshold variables respectively; $\gamma_1$, $\gamma_2$, and $\gamma_n$ are the specific threshold values; and I is the index function. When the conditions in parentheses are met, I is 1, otherwise, it is 0; control is the set of control variables (pd, ppi, and gov); and $\varepsilon_{it}$ is the random disturbance term.

*2.4. Data Sources*

Due to the long research period, in order to facilitate the analysis and research, the administrative division of 2015 is taken as the standard, and the data of each year are unified to the administrative unit of that year. The research data mainly includes five aspects: (a) China's provincial, municipal, and county-level administrative units are China's 1:250,000 basic geographic data provided by the Resources and Environment science data of the Chinese Academy of Sciences (http://www.resdc.cn/, accessed on 16 July 2020). Excluding Hong Kong, Macao, and Taiwan, there are 31 provincial-level administrative units and 2851 county-level units in the study area; (b) the registered population data of each county in the calculation of population urbanization come from the National Population Statistics of Counties and Cities of the People's Republic of China in 1995, 2000, 2005, and 2010. Since the data source only counted until 2012 and the division of "agricultural and non-agricultural" household registration was canceled in 2015, this paper used the "agricultural and non-agricultural" population data from 2014 to replace 2015. The data are from the official websites of statistical bureaus of all provinces, cities, and counties; (c) the land use data for 1995, 2000, 2005, 2010, and 2015 are obtained from the Data Center of Resources and Environmental Sciences, Chinese Academy of Sciences, with a resolution of 30 m (http://www.resdc.cn/); (d) the social statistics for regression are obtained from the statistical yearbook. (e) the stable night light data for economic urbanization calculation is provided by the NGDC website (https://ngdc.noaa.gov/eog/dmsp, last accessed on 16 July 2020), including 34 stable night light images without radiation calibration (1992–2013) from six DMSP satellites: F10, F12, F14, F15, F16, and F18. Background noises of the stable night data were identified and replaced with values of zero, and the final DN values for lit pixels ranged from 1 to 63. Since the data source is only updated to 2013, this paper uses the 2013 night light image to replace the 2015 data.

Due to the insufficient accuracy of the original noctilucent image (discontinuous problem and multi-sensor problem), we used the existing methods to correct it [56,57]. Firstly, the obtained global light images were extracted according to the Chinese regional boundaries, and then we obtained the Chinese light image data from 1992 to 2013. Secondly, the projection of light data was defined as the Lambert Azimuthal Equal projection, and data were resampled to a pixel size of 1 km. Thirdly, we chose "the method of invariant target region". The core idea of this method is to use a small area with a wide range of light values (0–63) but little change over the years to correct all images. Moreover, this method mainly consists of three steps: intercalibration, intra-annual composition, and inter-annual series correction. The specific operation is as follows: (a) Selection of the invariant target region. Of all the areas examined, Hegang City of Heilongjiang Province was selected as the "constant target area" due to its characteristics mentioned above. (b) Intercalibration. In order to solve the continuity problem, we chose the F16-2006 light image of Hegang City as the reference image and corrected all the images by constructing a quadratic equation of one variable. However, we found that the change in Hegang City from 1992 to 2013 was slightly larger, and the fitting degree of lighting value in 1992 and 2013 was only 0.78, less than 0.80. Therefore, after completing the whole process and comparing it with Lu, we thought that the coefficients provided by Lu might be more scientific, so we finally chose it (Table 2). (c) Intra-annual composition. Light data for some years will be provided by two sensors (F14-2000 and F15-2000; F15-2005 and F16-2005). In order to make full use of the images of the same year obtained by multiple sensors and ensure the stability and

continuity of night light image data, we averaged the two-image data of the same year. (d) Series correction. In order to solve the problem of individual pixel value mutation, we assumed that the DN value of a pixel in the DMSP/OLS lighting image should not be small in the following year DN value of the pixel in the previous year. According to the operation, we obtained the final Chinese lighting data set from 1992 to 2013.

**Table 2.** Parameters of a quadratic equation with one variable provided by Lu [79].

| Sensor | Year | a | b | c | $R^2$ |
|--------|------|------|------|------|------|
| F12 | 1995 | 0.034 | 0.513 | 0.485 | 0.851 |
| F14 | 2000 | 0.024 | 0.606 | 0.346 | 0.873 |
| F15 | 2000 | 0.028 | 0.578 | 0.485 | 0.878 |
| | 2005 | 0.026 | 0.872 | 0.123 | 0.855 |
| F16 | 2005 | 0.022 | 0.919 | 0.096 | 0.887 |
| F18 | 2010 | 0.021 | 0.510 | 0.901 | 0.851 |
| | 2013 | 0.016 | 0.271 | 0.680 | 0.850 |

## 3. Temporal and Spatial Pattern Evolution of County Urbanization in China

### 3.1. Characteristics of Spatio-Temporal Differentiation of Population-Land-Economy Urbanization

In order to describe the urbanization development pattern of China's counties in detail, the data of registered populations, nighttime lights, and land use were used to show the pattern of the registered population urbanization rate, economic urbanization rate, and land urbanization rate of 2851 counties in China during 1995–2015 (Figure 1).

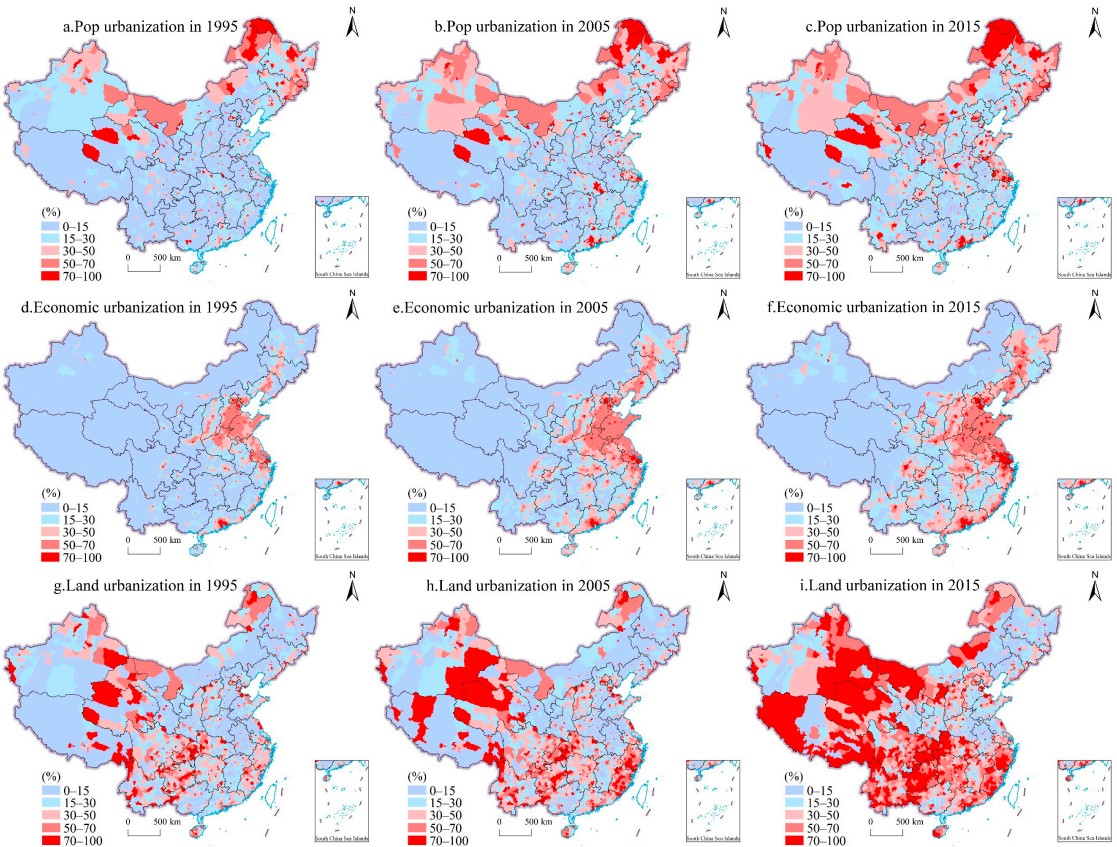

**Figure 1.** Spatial distribution of population-economy-land urbanization rates in China from 1995 to 2015.

County urbanization was represented by registered population data (Figure 1a–c). The average urbanization rate of the registered population of 2851 counties was 27.49% in 1995, 34.31% in 2005, and 38.62% in 2015, with an average annual growth rate of 2.19%. From the

perspective of spatial pattern, the urbanization rate in 1995 was generally low. Counties with high urbanization rates were mainly concentrated in Northern Xinjiang, border areas of Inner Mongolia, and Northeast China. The prototype of a high urbanization belt on the northern border had basically appeared. In addition, the eastern coastal areas of Beijing-Tianjin-Hebei, the Yangtze River Delta, and the Pearl River Delta also had small-scale high urbanization agglomeration areas with Beijing, Shanghai, and Guangzhou as the core. It could be seen that the dot high urbanization areas with provincial capitals and municipal districts as the core were also more prominent. In 2015, the Beijing-Tianjin-Hebei region, the Yangtze River Delta, and the Pearl River Delta along the eastern coast spread rapidly, forming a large area with a high urbanization rate. At the same time, high urbanization rate areas also appeared in the Shandong Peninsula, Jiangsu, Fujian, and other places. The high urbanization belt along the eastern coast had been basically formed, and a "herringbone" pattern had been formed with the high urbanization belt along the northern border, which is consistent with Liu's research results [21].

Economic urbanization was represented by light data (Figure 1d–f). The average urbanization rate of 2851 counties was 31.22% in 1995, 39.68% in 2005, 46.68% in 2015, and the average annual growth rate was 3.10%. From the perspective of spatial pattern, in 1995, the distribution of counties with high urbanization rate showed a pattern of "three points and one side", that was, ultra-high urbanization rate concentration points with Beijing, Shanghai, and Guangzhou as the core, and high urbanization rate areas along the eastern coast mainly with Shandong, Hebei, and Jiangsu. Moreover, the point regions with provincial capitals and municipal districts as the core were also areas with high economic urbanization rates. There was a significant difference in the rate of economic urbanization between the east and the west. In 2015, the difference in economic urbanization rate between east and west was more significant. In the east, three high-rate areas of Beijing-Tianjin-Hebei, Yangtze River Delta, and Pearl River Delta with Beijing, Shanghai, and Guangdong as the core had been formed. Furthermore, North China Plain was also a high-value concentration area of China's economic urbanization.

Land urbanization was represented by land-use data (Figure 1g–i). The average land urbanization rate of 2851 counties was 34.46% in 1995, 39.16% in 2005, and 50.37% in 2015, with an average annual growth rate of 2.63%. From the perspective of spatial pattern, in 1995, counties with a high land urbanization rate mainly were concentrated in the eastern coastal Yangtze River Delta, Fujian, Guangdong, and other places. Chongqing and Guizhou in southwest China formed concentrated and continuous high-value areas, and small high-value areas appeared in western Xinjiang, Qinghai, and Gansu. The difference between the north and the south in land urbanization was obvious. In 1995, the average land urbanization rate of counties was 35.51% in the east of the "Hu Line", while in the west, it was 28.49%, with a difference of 7.02%. However, the average land urbanization rate in the southern and northern counties was 38.68% and 29.90%, respectively, with a difference of 8.78%. In 2015, the gap between the north and the south in land urbanization rate is more significant, which is consistent with Gao's research results [34]. The difference in the average land urbanization rate between east and west counties of "Hu line" was 0.11%, and the difference between North and South was 18.58%. The southwest high-value area with Chongqing as the core was connected with the southeast coastal high-value area. Only some counties in western Jiangxi and central Guangxi had low land urbanization rates. The land urbanization rate of the county in the south had reached a high level, while the northeast plain and north China Plain in the north formed a wide range of low land urbanization rate regions.

In terms of spatial pattern, the urbanization of the registered population presents a "herring-shaped" pattern composed of high-value areas in the northern border area and high-value areas in the eastern coastal area, which further strengthens over time. The pattern of economic urbanization is "high in the east and low in the west". Land urbanization presents a pattern of "high in the south and low in the north", which is more obvious than the differentiation between the east and the west. In terms of development

level, the urbanization rate from highest to lowest is land, economy, and population. In terms of development speed, the urbanization rate from high to low is economy, land, and population. In general, the development level of population urbanization is the lowest, and the development rate is the slowest. However, urbanization is a development process with people as the core, so in the future, we must pay attention to the urbanization of "people".

### 3.2. Characteristics of Spatio-Temporal Differentiation of Comprehensive Urbanization

Based on the population, land, and economic development, a comprehensive urbanization rate index is constructed to show the spatio-temporal pattern of urbanization development at the county level in China (Figure 2). The spatial and temporal dynamics of the comprehensive urbanization level at the county level in China are significantly different, and the overall distribution still follows the "Hu Line". However, with the development, the comprehensive urbanization rate in western inland areas gradually increased. Three high-value areas of Beijing-Tianjin-Hebei, Yangtze River Delta, and Pearl River Delta, with Beijing, Shanghai, and Guangdong as the core have been formed along the eastern coast. BesidesIn addition, in 1995, the average comprehensive urbanization rate in 2851 counties was 31.06%. In 2015, it was 45.23%, with an average annual growth rate of 2.67%. It showed that the high-value areas with provincial capitals and municipal districts as the core form a multi-point distribution throughout the country.

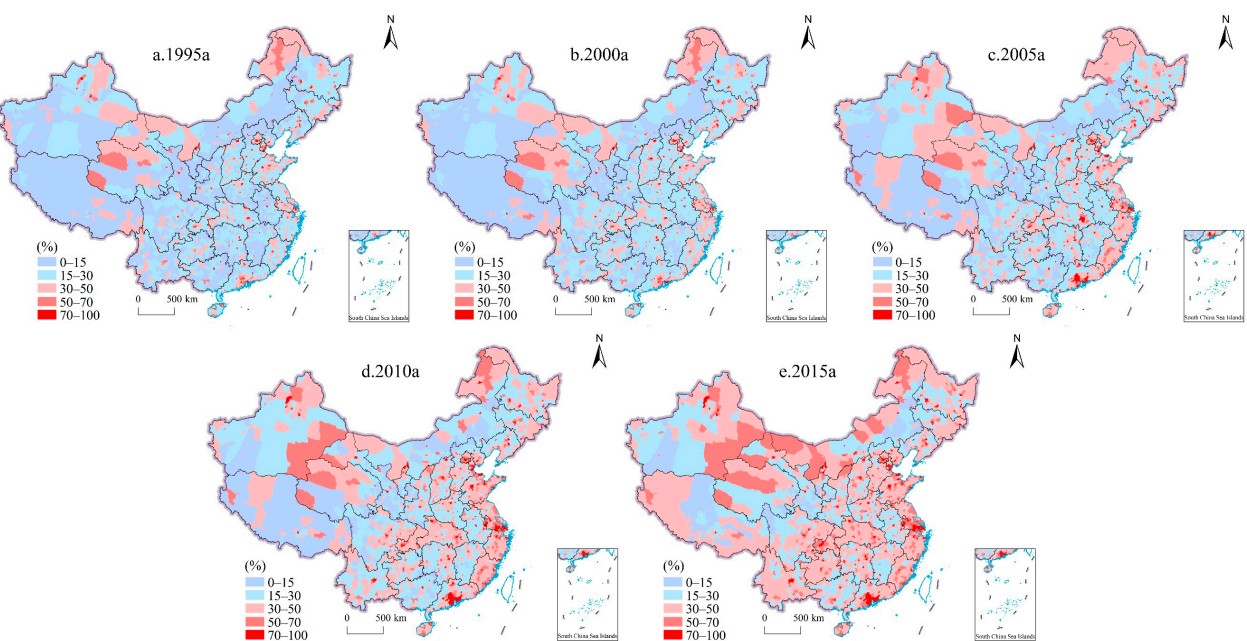

**Figure 2.** Spatial distribution of comprehensive urbanization rate of China's counties from 1995 to 2015.

In 1995, the overall comprehensive urbanization rate of counties in China was low. The average urbanization rate of 2851 counties was 31.06%, with obvious regional differences. A total of 41% of the 2851 counties in China had a comprehensive urbanization rate of 15–30%, followed by the total number of counties with an urbanization rate of 0–15% (685), indicating that China's county urbanization rate was extremely low. First, counties with a high urbanization rate were mainly concentrated in the eastern coastal areas, forming the high-value belt in the eastern coastal areas, among which Beijing, Shanghai, and Guangzhou were the three high-value points of urbanization rate. Furthermore, provincial capitals and municipal districts were the areas with high urbanization rates and scattered in points. There were also some counties in the western inland areas with relatively high comprehensive urbanization rates. Second, the areas with a low comprehensive urbanization rate were mainly concentrated in the western inland and other rough terrain areas, as well as non-provincial capitals and non-municipal districts.

In 2015, the average comprehensive urbanization rate of counties in China was 45.23%. Among the three core regions, the Pearl River Delta and Yangtze River Delta had achieved better development, followed by the Beijing-Tianjin-Hebei region. The urbanization rate in the inland areas of central and western China was growing rapidly, such as counties in Hubei, Chongqing, and Henan. The comprehensive urbanization rate of most counties was concentrated in the range of 30–50%. There were only 67 counties with a comprehensive urbanization rate of 0–15%, accounting for only 3% of the total number of counties in China. The number of counties with a comprehensive urbanization rate of 15–30% accounted for 22%. The number of counties with a comprehensive urbanization rate of 30–50% increased rapidly to 1291, accounting for 45% of the total number of counties in China. The number of counties with a comprehensive urbanization rate of 50–100% was also gradually increasing, accounting for 30%, indicating that China's county urbanization had reached the above medium level.

## 4. Results

### 4.1. Fixed Effects Panel Model Regression Results

After answering the first three questions mentioned above: "How high is the level of county urbanization in China? How to measure the level of urbanization? What are the characteristics of geographical imbalance?", we will further answer the question "What are the driving forces and mechanisms of county urbanization in China?". We selected four indicators to explain this problem, including population density, the proportion of primary industry in GDP, secondary and tertiary industrial structure, and the level of fiscal transfer payments. The regression results are shown in Table 3.

**Table 3.** Regression results of comprehensive urbanization.

|  | All | East | Mid | West |
|---|---|---|---|---|
| Pd | 6.42 *** | 5.97 *** | 2.76 *** | 9.31 *** |
|  | (0.58) | (0.99) | (0.91) | (0.97) |
| Ppi | −0.31 *** | −0.41 *** | −0.28 *** | −0.28 *** |
|  | (0.01) | (0.02) | (0.01) | (0.01) |
| Tos | 0.08 * | 1.19 *** | 0.13 * | −0.01 |
|  | (0.05) | (0.28) | (0.07) | (0.07) |
| Gov | 6.59 *** | 23.84 *** | 15.88 *** | 5.04 *** |
|  | (0.37) | (1.85) | (0.99) | (0.46) |
| Constant | 4.00 | 6.10 | 18.49 *** | −5.39 ** |
|  | (2.90) | (5.84) | (4.98) | (3.99) |
| F-value | 12.80 *** | 12.77 *** | 13.01 *** | 12.05 *** |
| $R^2$ | 0.37 | 0.44 | 0.43 | 0.38 |

Standard errors in parentheses; *** $p < 0.01$, ** $p < 0.05$, * $p < 0.1$

In order to explore the development mechanism of urbanization, the full sample is incorporated into the model and regression analysis is conducted on each factor. At the same time, there is a large difference among different counties with strong heterogeneity. Therefore, the counties are divided into eastern, central, and western regions according to their geographical location, and the regression results are shown in Table 3. Overall, a large population scale, reasonable industrial structure, and strong government capacity play a good role in promoting the development of county urbanization. Specifically, (a) When all counties are included in the model, the regression coefficients of pd and gov are 6.42 ($p < 0.01$) and 6.59 ($p < 0.01$), respectively, indicating that population size and government capacity have promoted the improvement of the county urbanization rate. In terms of industrial structure, the regression coefficient of ppi is −0.31 ($p < 0.01$), indicating that the development of primary industry may inhibit the improvement of county urbanization rate. However, the regression coefficient of Tos is 0.08 ($p < 0.1$), indicating that the tertiary industry could better promote the development of urbanization than the secondary industry. (b) In the eastern region, the regression results are similar

to those at the national level. The regression coefficients of pd, ppi, Tos, and gov are 5.97 ($p < 0.01$), −0.41 ($p < 0.01$), 1.19 ($p < 0.01$), and 23.84 ($p < 0.01$), respectively. Among them, Tos has a stronger significance and a larger regression coefficient, indicating that the tertiary industry plays a greater role in improving the urbanization rate in eastern China. (c) In central China, the regression coefficients of pd, proportion of primary and secondary industries, proportion of secondary and tertiary industries, and government capacity were 2.76 ($p < 0.01$), −0.28 ($p < 0.01$), 0.13 ($p < 0.1$), and 15.88 ($p < 0.01$), respectively. (d) In the western region, Tos shows particularity, and its regression coefficient is −0.01, which does not pass the significance test. It shows that the development of the tertiary industry cannot contribute to the improvement of county urbanization rate in western inland areas, while the development of the secondary industry may be more conducive to the development of urbanization.

China is a vast country with complex and diverse landforms. Different types of county urbanization may have different driving mechanisms. Therefore, according to the terrain and geomorphology, the counties are divided into plain counties, hill counties, and mountain counties, and regression is conducted respectively (Table 4). Whether in a plain county, hill county, or mountain county, the population scale and government capacity can promote the urbanization rate of the county, while the development of the primary industry will limit the urbanization development. However, Tos shows differences among the three types. In plain counties, the Tos regression coefficient is 0.07, but it does not pass the significance test, indicating that its promotion effect on the urbanization rate is not obvious. In hilly counties, Tos regression coefficient is −0.45 ($p < 0.01$), indicating that it inhibits the development of urbanization. It shows that the development of the tertiary industry cannot contribute to the improvement of the county urbanization rate in hill countries, while the development of the secondary industry may be more conducive to the development of urbanization. In mountainous counties, Tos regression coefficient is 0.12 ($p < 0.1$), indicating that the development of the tertiary industry will significantly promote the improvement of the urbanization rate.

**Table 4.** Regression results of different types of areas.

|  | **Plain County** | **Hill County** | **Mountain County** |
|---|---|---|---|
| Pd | 6.86 *** | 6.82 *** | 6.71 *** |
|  | (0.99) | (1.17) | (0.90) |
| Ppi | −0.34 *** | −0.27 *** | −0.30 *** |
|  | (0.01) | (0.01) | (0.01) |
| Tos | 0.07 | −0.45 *** | 0.12 * |
|  | (0.08) | (0.15) | (0.07) |
| Gov | 9.23 *** | 18.73 *** | 5.61 *** |
|  | (1.27) | (1.42) | (0.42) |
| Constant | 0.33 | −0.80 | 4.28 |
|  | (5.60) | (6.09) | (3.96) |
| F-value | 11.48 *** | 13.39 *** | 13.08 *** |
| $R^2$ | 0.37 | 0.37 | 0.40 |

Standard errors in parentheses; *** $p < 0.01$, ** $p < 0.05$, * $p < 0.1$.

Finally, based on t from the official documents published by each Chinese province, we divide the counties into different functional zones, including key development zones, optimized development zones, main grain-producing areas, and ecological protection zones, and then carry out regression for the four types (Table 5). In the four subregions, both population size and government capacity play a significant positive role in the improvement of urbanization, while the development of the primary industry significantly inhibits the improvement of the urbanization rate. However, Tos shows great differences in different functional zones, specifically: in the key development zone and ecological protection area; Tos regression coefficients are 0.50 and 0.03, respectively, which both fail the significance test, indicating that Tos has no significant promoting effect on urbanization. In the optimized

development zone, the Tos regression coefficient is 5.66, and through the significance test of 1%, which shows that the good development of the tertiary industry will better promote the increase of urban rate. In the main grain-producing areas, Tos also shows a positive effect, and its regression coefficient is 0.14 ($p < 0.1$).

**Table 5.** Regression results of different functional zones.

| | Key Development Zone | Optimized Development Zone | Major Grain-Producing Zone | Ecologic Protection Zone |
|---|---|---|---|---|
| Pd | 9.09 *** | 12.30 *** | 5.78 *** | 4.70 *** |
| | (1.28) | (2.45) | (0.89) | (0.97) |
| Ppi | −0.46 *** | −0.62 *** | −0.28 *** | −0.25 *** |
| | (0.02) | (0.09) | (0.01) | (0.01) |
| Tos | 0.50 | 5.66 *** | 0.14 * | 0.03 |
| | (0.34) | (1.67) | (0.08) | (0.06) |
| Gov | 14.97 *** | 27.40 * | 13.35 *** | 5.45 *** |
| | (1.74) | (15.50) | (0.84) | (0.43) |
| Constant | −9.71 | −29.95 * | 2.77 | 12.46 ** |
| | (7.31) | (16.00) | (4.81) | (3.92) |
| F-value | 12.16 *** | 9.75 *** | 10.95 *** | 13.34 *** |
| $R^2$ | 0.46 | 0.47 | 0.44 | 0.33 |

Standard errors in parentheses; *** $p < 0.01$, ** $p < 0.05$, * $p < 0.1$.

In conclusion, a large population size and strong government capacity will promote the rapid improvement of county urbanization rate in China, but a high proportion of primary industry will inhibit the development of urbanization. In different types of counties, the performance of Tos is quite different, which shows that relying solely on the tertiary industry or the secondary industry cannot promote the development of urbanization effectively. Each county should adjust its industrial structure reasonably according to local conditions. For example, an optimized development zone should vigorously develop the tertiary industry, while the development of the secondary industry in hilly counties may be more conducive to the development of urbanization.

### 4.2. Threshold Model Regression Results

The empirical results of the panel model have shown that there are great differences in the impact of the Tos on urbanization in different regions, which preliminarily shows that the tertiary industry plays a stronger role in promoting the urbanization rate in developed regions, while the secondary industry plays a greater role in relatively developing regions. However, can the regression results of different regions show that with the increase in urbanization rate, the promotion effect of the tertiary industry increases while the promotion effect of the secondary industry decreases? This paper then uses the threshold model to test the threshold effect of Tos on comprehensive urbanization by taking the population urbanization rate, land urbanization rate, and economic urbanization rate as threshold variables.

In Table 6, when population urbanization is used as the threshold variable, within the 95% confidence interval, the whole sample passes the double threshold test, and the double threshold values are 38.56% and 58.73%, respectively; when land urbanization is used as the threshold variable, within the 95% confidence interval, the whole sample passes the double threshold test, and the double threshold values are 0.18% and 38.14%, respectively; when economic urbanization is used as the threshold variable, within the 95% confidence interval, the whole sample passes the double threshold test, and the double threshold values are 32.27% and 59.06%, respectively. Based on this, this paper takes population urbanization (Model 1), land urbanization (Model 2), and economic urbanization (Model 3) as threshold variables for regression, and the estimation results are shown in Table 7.

**Table 6.** Threshold estimates and test results.

| Threshold Variable | Threshold Test | Threshold Value | *p* Value | 95% Confidence Interval |
|---|---|---|---|---|
| Pop_urban | Single threshold test | 38.56 | 0.00 | [38.135, 38.958] |
| | Double threshold test | 58.73 | 0.00 | [57.657, 60.185] |
| | Triple threshold test | — | — | — |
| Land_urban | Single threshold test | 0.18 | 0.00 | [0.000, 1.851] |
| | Double threshold test | 38.14 | 0.00 | [37.621, 38.313] |
| | Triple threshold test | — | — | — |
| Eco_urban | Single threshold test | 32.27 | 0.00 | [29.369, 32.430] |
| | Double threshold test | 59.06 | 0.00 | [58.617, 60.480] |
| | Triple threshold test | — | — | — |

—indicates that there is no triple threshold.

**Table 7.** Threshold panel model estimation results.

| Variable | Model 1 | Model 2 | Model 3 |
|---|---|---|---|
| Tos ($\gamma_1$) | −0.02 | −0.45 *** | −0.21 *** |
| | (0.05) | (0.08) | (0.05) |
| Tos ($\gamma_2$) | 2.55 *** | −1.24 *** | 0.74 *** |
| | (0.20) | (0.10) | (0.09) |
| Tos ($\gamma_3$) | 6.49 *** | 0.84 *** | 4.75 *** |
| | (0.34) | (0.06) | (0.30) |
| Pd | 6.79 *** | 6.00 *** | 5.13 *** |
| | (0.57) | (0.57) | (0.58) |
| Ppi | −0.31 *** | −0.29 *** | −0.30 *** |
| | (0.01) | (0.01) | (0.01) |
| Gov | 6.19 *** | 6.48 *** | 6.61 *** |
| | (0.36) | (0.36) | (0.36) |
| _cons | 1.73 | 5.98 *** | 9.60 *** |
| | (2.85) | (2.84) | (2.89) |
| F-test | 11.16 *** | 11.20 *** | 12.16 *** |
| _cons | 1.73 | 5.98 *** | 9.60 *** |

Standard errors in parentheses; *** $p < 0.01$, ** $p < 0.05$, * $p < 0.1$.

According to the results of the three models, there is an obvious nonlinear relationship between the Tos and the comprehensive urbanization in the process of urbanization. In the early and middle stages of urbanization, the secondary industry plays a strong role in promoting the urbanization rate. With the development, the role of the secondary industry is gradually weakened, and the tertiary industry occupies a dominant position. Specifically: (a) Taking population urbanization as the threshold variable, when it is at the first threshold (pop_urban $\leq$ 38.56%), the Tos regression coefficient is −0.02, indicating that the urbanization rate will not increase with the increase of Tos, and the indicator Tos has a negative correlation with the secondary industry and a positive correlation with the tertiary industry, so the secondary industry has an obvious effect on the urbanization rate; when it is in the second threshold range (38.56% < pop_urabn $\leq$ 58.73%), the Tos regression coefficient is 2.55 ($p < 0.01$), indicating that the comprehensive urbanization rate will increase with the increase of Tos, and the promotion effect of the tertiary industry will gradually exceed that of the secondary industry; when crossing the second threshold (58.73% $\leq$ pop_urban), Tos is still positively correlated with the comprehensive urbanization rate (the correlation coefficient is 6.49), and the tertiary industry promotes the urbanization rate more obviously. (b) Taking land urbanization as the threshold variable, when it is in the first threshold interval (land_urban $\leq$ 0.18%) and the second threshold interval (0.18% < land_urabn $\leq$ 38.14%), the Tos regression coefficients are −0.45 ($p < 0.01$) and −1.24 ($p < 0.01$) respectively, indicating that there is an obvious negative correlation between Tos and the comprehensive urbanization rate, and the development of the secondary industry will greatly promote the improvement of the comprehensive urbanization rate; when crossing the second threshold (38.14% $\leq$ land_urban), the regression coefficient of Tos is

0.84 ($p < 0.01$). The comprehensive urbanization rate will increase with the increase of Tos, and the tertiary industry will begin to play a stronger role in promoting it. (c) Taking economic urbanization as the threshold variable, when it is at the first threshold (eco_urban $\leq 32.27\%$), the Tos regression coefficient is $-0.21$ ($p < 0.01$), indicating that the secondary industry can better promote the improvement of the comprehensive urbanization rate; when it is located in the second threshold range (32.27% < eco_urban $\leq$ 59.06%) and crosses the second threshold (59.06% $\leq$ eco_urban), the Tos regression coefficients are 0.74 ($p < 0.01$) and 4.75 ($p < 0.01$) respectively, indicating that the role of the tertiary industry in promoting the urbanization rate has exceeded that of the secondary industry.

Generally, when the urbanization rate is lower than 38%, the role of the secondary industry in promoting the urbanization rate is very obvious, and the role of the tertiary industry is weak; when the urbanization rate exceeds 38%, the role of the tertiary industry is gradually enhanced, while the role of the secondary industry is gradually weakened; when the urbanization rate exceeds 59%, the role of the tertiary industry in promoting the urbanization rate is more obvious, and its role is far more than that of the secondary industry.

## 5. Discussion

This study comprehensively reveals the temporal and spatial dynamic pattern of county urbanization in China from 1995 to 2015. We answered four questions: "How to measure the level of urbanization?"; "How high is the level of county urbanization in China?"; "What are the characteristics of geographical imbalance?"; and "What is the driving mechanism of urbanization?".

### 5.1. The Rationality of Index Construction

The comprehensive urbanization index we constructed can more scientifically measure the real urbanization development level. On the one hand, closely adhering to the definition of urbanization, we combined multi-dimensional (population, land, and economy) and multi-source data (population statistics, land use, and night light), which solved the limitations of single indicators [22,79] and the traditional attribute of comprehensive indicators [80] in the existing literature, and finally provided a new urbanization measurement index. On the other hand, in order to test the objectivity of comprehensive urbanization constructed, the indexes of permanent resident population urbanization [81,82] commonly used in scholar and government statistics are selected and compared with comprehensive urbanization to reflect the differences between the two measurement methods.

In 2000, the average permanent resident urbanization rate of China's 2851 counties was 37.80 %. In 2010, it was 46.65%, with an average annual growth rate of 4.03%. According to the comprehensive urbanization rate measurement, the average urbanization rate of 2851 counties was only 33.54% in 2000, 41.39% in 2010, and the average annual growth rate was 2.80%. As can be seen from Figure 3, the urbanization rate of permanent residents was higher than the comprehensive urbanization rate in both 2000 and 2010. At the same time, in the permanent population statistics, the northwest border area belonged to the area with a high urbanization rate. However, in the comprehensive urbanization based on the combination of population, land and economy, the urbanization rate of the northwest border region declined to a certain extent, and the "false high urbanization error" in the northwest border region was reduced. In addition, the comprehensive urbanization measurement method highlighted the core status of the Beijing-Tianjin-Hebei region, the Yangtze River Delta, and the Pearl River Delta.

In order to make a more detailed distinction between the two types of urbanization, the urbanization rate is divided into five intervals, and the number of counties in each interval is counted (Figure 4). In 2000, the number of counties with a resident population urbanization rate of 15–30% was 884, accounting for 31% of the number of counties in China; the number of counties with a comprehensive urbanization rate of 15–30% was 1246, accounting for 44% of the number of countries in China. The number of counties

in the urbanization rate of 70–100% also had a large difference between the two calibers. Among them, the number of counties in the ordinary population statistics caliber and the comprehensive caliber were respectively 573 and 306. This showed that the urbanization rate of permanent residents in 2000 was higher than the comprehensive urbanization rate. In 2010, the number of counties with a resident population urbanization and a comprehensive urbanization rate of 30–50% were respectively 1072 (accounting for 37.6%) and 1016 (accounting for 35.6%). However, the number of comprehensive urbanization counties with an urbanization rate in the range of 50–100% was 750, which was less than 936 counties with a permanent population. It further indicated that the urbanization rate of a permanent resident population overestimated the urbanization process at the county level in China.

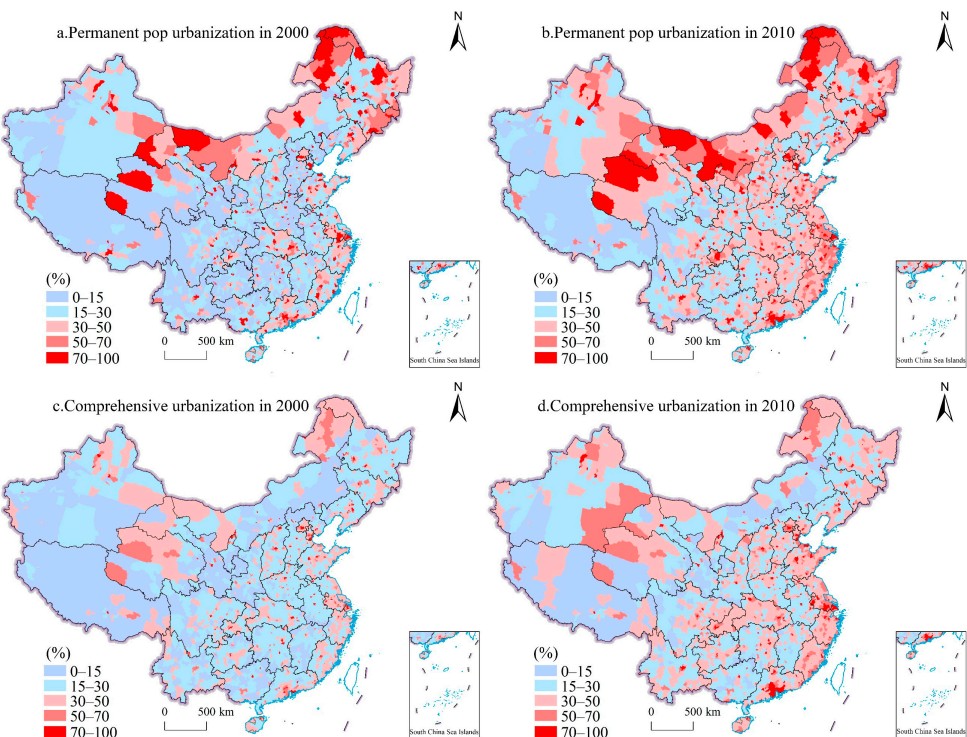

**Figure 3.** Comparison between permanent resident population urbanization and comprehensive urbanization.

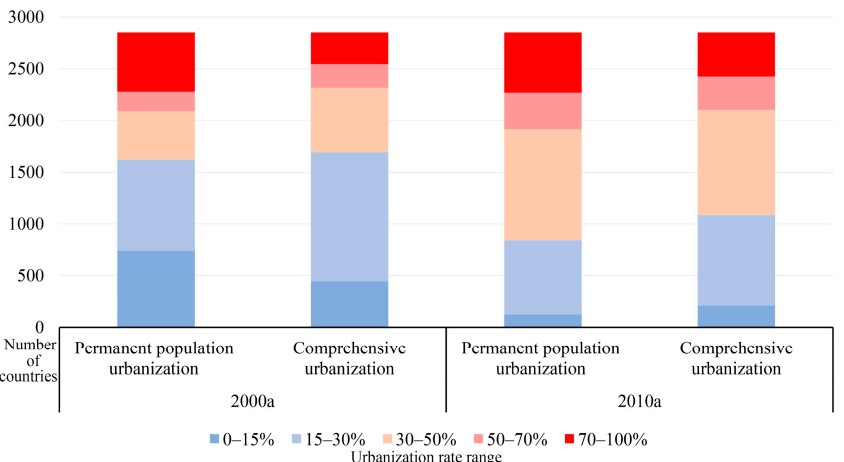

**Figure 4.** Number of counties with permanent population urbanization and comprehensive urbanization in different urbanization rates.

The measurement results of the two methods are similar. However, the comparison results show the urbanization level of the permanent population may overestimate China's urbanization process, which is also consistent with existing research results [14,15,27].

*5.2. The High-Quality Development of Urbanization*

First, narrowing regional differences is the primary task for achieving high-quality urbanization development. Our study found that there are obvious regional differences in population, economy, land, and comprehensive urbanization [83,84]. Although China has adopted regional strategies (such as the western development strategy and rising strategy in the central region), the regional gap is still large. On the basis of the difference between east and west, the widening of the difference between north and south has become a new regional development problem in China [85,86]. In the future, in-depth analysis of the formation mechanism of regional differences is an important research direction.

Second, "people" urbanization should be the core of high-quality urbanization, but it is still underestimated. By measuring the rate of population, economy, and land urbanization, respectively, it is found that the rate of population urbanization is lower than that of economic urbanization and land urbanization both in terms of development level and speed. Overemphasis on economic and land development and neglect of population urbanization will easily cause great pressure on the environment [87] and even widen the gap between the rich and the poor [17,66]. In the future, the government should continue to firmly promote the "three hundred million" policy, promote the settlement of 100 million agricultural transfer population in cities and towns, improve the living conditions of 100 million people, and guide 100 million people to urbanize nearby in the central and western regions.

Third, the optimization of industrial structure is one of the necessary ways to realize high-quality urbanization. The government promoted the development of early urbanization [88,89]. With economic development and marketization, the government's role in promoting urbanization has gradually weakened, and county economic development and industrial structure have become important factors for the improvement of urbanization. At the same time, there is an obvious nonlinear relationship between the impact of Tos on urbanization, and 38% and 59% can be used as important nodes of urbanization [90]. Each county at different stages of development should adjust its industrial structure reasonably according to local conditions. Therefore, in the early and middle stages of urbanization, we should vigorously develop the secondary industry; with the increase in urbanization rate, counties should gradually pay attention to the development of the tertiary industry [91,92].

*5.3. Application of Night Light Data*

Due to the close relationship between lighting data and economic development [69,70], light data provided a valuable data source for elucidating the dynamics of China's urbanization [71–73,93]. At the same time, due to the attributes of a long time scale and high spatial resolution, light data can not only estimate the urbanization development at the national, regional, and city scales but also play a role at the county and grid scales. For example, Amaral [94] used DMSP/OLS data to estimate the population of the Amazon region and realized the population estimation in the areas lacking statistical data, which showed that light data has strong application at the regional scale; Yu [95] used light data to build a 500-m population grid, proving that light data can also be applied on a microscopic scale. Through the literature review, the research on using light to represent urbanization is mainly divided into two aspects. On the one hand, based on the area attribute of light data, by distinguishing between bright cities and dark villages, the information on the urban boundary is extracted [96], and finally, the study of urbanization is realized. For example, Yang [97] proposed a method to measure land urbanization level based on DMSP/OLS nighttime light data of long time series and takes the Bohai Rim region as an example to study the spatio-temporal measurement of land urbanization level from 1992 to 2010. Liu [98] explored the urbanization process of Shanxi, Shaanxi, and Mongolia by using the

area attribute of lighting. On the other hand, the intensity attribute of light data is used to reflect the urbanization process. For example, Gao [76] constructed urbanization indicators through lighting brightness attributes and analyzed the dynamics of urbanization levels in China from 1992 to 2012. However, due to the simple information contained in light data, it cannot fully reflect the information of the research object, and its advantages cannot be fully utilized, but multi-source data fusion can solve these problems [99]. Therefore, we use multi-source data fusion for research, which not only makes full use of the advantages of lighting data but also can more comprehensively and scientifically reflect the process of urbanization in China.

Although light data has advantages in reflecting urbanization development, the shortcomings of the data weaken the accuracy of the nighttime light data in quantitative analysis. The first problem is the time scale of the data. Although DMSP/OLS provided a long time data set, it was only updated to 2013; NPP/VIIRS data are dated, but the earliest data date is only 2012, which makes it difficult to carry out research on a long time scale. The second problem is the spatial scale of data. The grid size of DMSP/OLS data is about 1 km, while the grid size of NPP/OLS data is about 500 m, so the fusion of the two kinds of data is limited to some extent. At the same time, DMSP/OLS with a 1 km grid size will have large errors when studying small scales. The third problem is the brightness value of the light. Due to sensor problems, the maximum value of DMSP/OLS data is only 63, which will lead to large errors in estimating economic and urbanization development.

In the future, solving the problems of lighting data is a research focus. The first is the fusion of DMSP/OLS data and NPP/VIIRS data. In order to construct light data at a longer time scale, it is of great significance to combine the two data sources scientifically and effectively in future research on urbanization. The second is light data correction. A good solution to the data quality problem (the value range of DMSP/OLS data) can reduce the error in the application research. The third is the fusion of multi-source data. The nighttime light data information is relatively simple. Compared with single lighting data, the fused data can give full play to the respective advantages of multi-source data (such as statistical data and lighting data, vector data and lighting data) to expand the application scenarios of data.

### 5.4. Limitations and Future Work

This study still had some limitations. Firstly, due to the limitation of data availability, the time period for this study was from 1995 to 2015, and the development of county urbanization in China after 2015 was not measured and analyzed. Secondly, in the process of using social statistics, night light, and land use remote sensing data to construct comprehensive urbanization indicators at the county level in China, the weights of all three are set at 1/3, which may cause errors in some regions. Thirdly, because of the difficulty of obtaining population data, this paper uses registered residence populations to construct a comprehensive urbanization rate. However, the number of floating populations in China has increased over the past years, and the use of registered residence populations may cause some errors. Further, in addition to the transformation of population, land, and economy, the transformation of rural culture and urban culture is also an important symbol of the process of urbanization. Limited by the data, we do not measure the transformation of cultural elements. Finally, since DMSP lighting data is only updated to 2013, we used 2013 data instead of 2015, which would also cause some errors. In future research, the extension of the time scale, the acquisition of population data and the scientific determination of weight are the key points. For the measurement of urbanization, incorporating more elements such as culture and society into the model will more comprehensively and truly reflect the process of urbanization. For lighting data, it is also one of the key points to integrate DMSP data and NPP data more scientifically to form a set of longer time data sets and apply it to the research of social and economic activities.

## 6. Conclusions

In order to measure and characterize the county urbanization in China from 1995 to 2015 in detail, this paper constructs a comprehensive urbanization index by fusing the demographic statistics, nighttime lights data, and land use remote sensing data. Moreover, we use the regression model to analyze the factors affecting the county urbanization level. Finally, the following conclusions could be drawn:

(a) In 1995, the average urbanization rate of 2851 counties of the registered population urbanization rate, economic urbanization rate, and land urbanization rate was 27.49%, 31.22%, and 34.46% respectively, and in 2015, they reached 38.62%, 46.68%, and 50.37%, respectively. The urbanization of registered populations presented a "herring-shaped" pattern consisting of the high-value areas on the northern border and the high-value areas in the eastern coastal areas, and it would further strengthen as time went on; economic urbanization presented a pattern of "high in the east and low in the west"; and land urbanization presented a pattern of "high in the south and low in the north", which was more obvious than the differentiation between the east and the west. In terms of development level, population urbanization rate was lower than economic urbanization rate, while economic urbanization rate was lower than land urbanization rate. In terms of development speed, the economic urbanization rate was faster than the land urbanization rate, and the land urbanization rate was faster than the population urbanization rate.

(b) By integrating the three elements of population, economy, and land, the comprehensive urbanization rate was constructed and calculated. The average comprehensive urbanization rate of 2851 counties in China was 31.06% in 1995 and 45.23% in 2015. The spatial and temporal dynamics of comprehensive urbanization level at the county level in China were significantly different, and the overall distribution still followed the "Hu Line". The comprehensive urbanization rate east of the line was higher, while the comprehensive urbanization rate west of the line was lower. However, with the passage of time, the comprehensive urbanization rate in western inland areas gradually increased. Counties with high urbanization rates were mainly concentrated in the eastern coastal areas, forming a high-value belt along the eastern coastal areas. Beijing, Shanghai, and Guangzhou were the top three.

(c) The urbanization rate of permanent resident populations overestimates the urbanization process at the county level in China. In 2000, the average permanent urbanization rate of 2851 counties was 37.80%, and the comprehensive urbanization rate was 33.54%. In 2010, the two types of urbanization rates were 46.65% and 41.39%, respectively.

(d) Large population size, reasonable industrial structure, and strong government capacity can play a positive role in promoting urbanization. In the industrial structure, the primary industry will limit the development of urbanization, but the promotion effect of secondary and tertiary industries on urbanization is different in different counties

(e) There is an obvious nonlinear relationship between urbanization rate and Tos. When the urbanization rate is lower than 38%, the role of the secondary industry in promoting the urbanization rate is obvious; when the urbanization rate exceeds 38%, the role of the tertiary industry is gradually enhanced, and the secondary industry is gradually weakened; when the urbanization rate exceeds 59%, the role of the tertiary industry is more obvious and far more than that of the secondary industry.

Through the research of this paper, we found it was feasible to use the multi-source data fusion of demographic statistics, night lights, and land use remote sensing to construct a comprehensive urbanization index, as well as meaningful to comprehensively and truly reflect China's urbanization at the county scale and guide the future development of urbanization.

**Author Contributions:** Conceptualization, B.Z. and C.M.; methodology, B.Z. and C.M.; software, B.Z. and J.Z.; validation, B.Z. and C.M.; formal analysis, B.Z. and J.Z.; resources, B.Z. and C.M.; data curation, B.Z. and C.M.; writing—original draft preparation, B.Z. and C.M.; writing—review and editing, B.Z., C.M. and J.Z.; visualization, B.Z., C.M. and J.Z.; supervision, C.M.; project administra-

tion, C.M.; funding acquisition, C.M. All authors have read and agreed to the published version of the manuscript.

**Funding:** This research was funded by the National Natural Science Foundation of China (Fund-project host: Miao Changhong), grant number 42171186; The Major Project of China National Social Science Fund in Art (Fundation sub-project host: Miao Changhong), grant number 21ZD03.

**Data Availability Statement:** (a) China's provincial, municipal, and county-level administrative units are China's 1:250,000 basic geographic data provided by the Resources and Environment science data of the Chinese Academy of Sciences at http://www.resdc.cn/; (b) The registered population data of each county in the calculation of population urbanization come from the National Population Statistics of Counties and Cities of the People's Republic of China in 1995, 2000, 2005 and 2010 at https://data.cnki.net/yearbook; (c) The land use data of 1995, 2000, 2005, 2010 and 2015 are obtained from the Data Center of Resources and Environmental Sciences, Chinese Academy of Sciences, with a resolution of 30 m at http://www.resdc.cn/; (d) The stable night light data for economic urbanization calculation is provided by NGDC website at https://ngdc.noaa.gov/eog/dmsp; (e) The social statistics for regression are obtained from statistical yearbook at https://data.cnki.net/yearbook. All data accessed on 16 July 2020.

**Conflicts of Interest:** The authors declare no conflict of interest.

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
