# Peer review of "Urbanization Level in Chinese Counties: Imbalance Pattern and Driving Force"

_remotesensing, doi:10.3390/rs14092268_

Round 1

Reviewer 1 Report

Review report for the paper “Urbanization level in Chinese counties: imbalance pattern and driving force”.

Significance :

. The scientific content of this paper is correct. 
. The technical quality of this paper is correct. 
. The conclusion is correctly justified and but it should be better supported by the results. 
. The limits of the results obtained in this paper are not mentioned. This point should be presented. 
Quality of presentation:

. The abstract is not clear and it doesn’t presents correctly the subject addressed in this paper. 
. Introduction - What are your contributions?

. Why did you use GIS regression model in this research? Why not other GIS tools? You should discuss advantages of this tool over other available in the field.

. Better highlight novelty in the study.

. Better define motivations for the research.

. Why do we need this study?

. The data and analyses should be better presented. Add more discussion on the results. Add comparisons with existing approaches.

. Literature review is not well prepared. Based on LR you should define gap you are trying to cover.  

 . Generally, validation and comparisons of the results is well prepared. But, more discussion in on the results of the case study are needed. The authors need to discuss these values and the performance of their approach. How should we know about the quality of these solutions? Could you compare these results with more existing approaches in literature? The improvement must be discussed.

. Comparisons with the results of other tools should be presented.

. The conclusion section seems to rush to the end. The authors will have to demonstrate the impact and insights of the research. The authors need to clearly provide several solid future research directions. Clearly state your unique research contributions in the conclusion section. Add limitations of the model.  
Scientific soundness :
. The subject addressed in this paper is relevant.  
Interest to the readers :
. In my opinion, method of this paper seem to be interesting for the readership of the journal. 

Author Response

Dear experts,

Thank you for your valuable comments, we have made detailed revisions based on your comments.

Reviewer 2 Report

The paper deals with an interesting theme highly topical in the context of the current World research trends. The authors focused to present the dynamic changes of Chinese urbanization in population, land, economy and their comprehensive level in 2851 counties from 1995 to 2015 by using household census data, remote sensing data of land use and nighttime light data and constructing a composite urbanization index, and investigate the driving forces by regression model. I appreciate that authors applied method of urbanization measurement . especially fixed effects panel regression model and also other research methods and data sources as well as the interpretation of the obtained results.

The title of the paper is acceptable and adequate and no major changes are necessary. I find the abstract acceptable and well structured. The manuscript has a sufficient scientific value and the information provided represents widening of knowledge. The conclusions are based entirely on the results and the methods used are adequate. The relation between the scientific value and the extent is acceptable. The language and style of the text are at an acceptable level. The tables and illustrations used in the paper are adequate; however I consider the number of references incomplete. The topic dealt with in the paper is also covered by other authors in papers.

I recommend language correction of the text by a native speaker, if possible. I have no other remarks of a rather significant nature concerning the paper. The results are valuable and the scientific paper brings new original data. The manuscript is acceptable after minor revision with minor amendments required; no re-review is necessary. I recommend the paper for the print.

So no elements that should be corrected:

Conclusion: any limitation of your research? So please add it.

I recommend amending the references. This issue is also covered by the newer papers from other authors. I recommend adding some papers into the references.

Figure 3 – text is small (especially text of legends).

As you see, there is not too much to correct according to my opinion.

Good luck in your future scientific work.

Author Response

(The authors gave the same response as above.)

Reviewer 3 Report

The manuscript entitled “Urbanization level in Chinese counties: imbalance pattern and driving force” constructed population, land, economic, and comprehensive urbanization metrics respectively based on the integration of multiple sources of data. According to these indicators, the urbanization characteristics of Chinese counties and their driving factors were analyzed. It is meaningful that the manuscript is able to depict the characteristics of China's urbanization process more comprehensively and accurately through the constructed of a comprehensive urbanization index. However, the manuscript still has many flaws that concerned me! My main concern is the weak connections between the content of the manuscript and the topic of "remote sensing"! Yes, the manuscript does use remote sensing data, but I think it is insufficient.

In addition, there are some more specific comments about the manuscript:

  1. L110: “with my country’s national conditions.” Please revise it.
  2. L117-118: These consecutive interrogative sentences need to be followed by their meaning or measures.
  3. L127: Please delete "years"
  4. L150: Does Popall refer to the total registered population? If yes, please define it clearly.
  5. L158-161: First, Equation 2 does not seem to be consistent with the previous definition. Second, ul represents urban land area means what?
  6. L174: How is AreaN defined? What is the threshold of the lighted area?
  7. L179-187: Please simplify this part.
  8. L214-218: Please check carefully that the variables in the formula should be clearly explained.
  9. L231: “i stands for the country”?
  10. L237: “pd, ppi and fiscal”: These variables need to be consistent with the ones mentioned earlier.
  11. L308: In fact, from the figure, it seems that the east-west gap is more obvious than the north-south one, especially in 1995.
  12. L352: Qinghai has quite a few higher urbanization rate units.
  13. L384: First, I think it is unfair to compare the comprehensive urbanization index with the permanent population urbanization index as the former integrates information on three dimensions while the latter only has information on population. Therefore, it is more appropriate to directly compare the urbanization rates of the permanent and registered populations. The manuscript emphasizes that the comprehensive index reduces the "false high urbanization error" caused by the permanent population index. However, the fact remains that this error is still serious in the results of the registered population index. It may even be worse than the permanent population index. In fact, the massive population movement caused by reform and development cannot be ignored and is a fact of life. These populations live permanently in the cities and engage in non-agricultural activities. From my point of view, the resident population is more accurate than the registered population. This may also be why the distinction between agricultural and non-agricultural registered population has been abolished.
  14. L405-406: How did the authors conclude that the resident population index is an overestimate and not the registered population is an underestimate? The criteria for the determination need to be given.
  15. The titles of Tables 2 and 3 are incorrect.
  16. L464-466: What is the basis for the classification?
  17. L519: “and the secondary industry has an obvious effect on the urbanization rate” The conclusion of this sentence needs a better explanation. A similar question follows.
  18. L633-634: This sentence does not seem to be related to this paper, please consider deleting it.

Author Response

(The authors gave the same response as above.)

Round 2

Reviewer 1 Report

The authors have addressed the point of my concern. I am happy with their corrections. Hence, I would like to recommend this manuscript to be published.

Author Response

Dear expert,

Thank you for your constructive comments, we have learned a lot. At the suggestion of the academic editor, we have revised some details again.  Good luck in your future scientific work.

Best wishes.

Reviewer 3 Report

The manuscript was improved a lot after addressing the suggestions of the reviewers. I also appreciate the author's patience in answering my questions, even though some of them are still not perfectly solved. However, I still think it can now be considered for publication in a journal.

Author Response

(The authors gave the same response as above.)
